# Clinical and Epidemiological Assessment of Children and Adolescents Hospitalized with SARS-CoV-2 in the Pre-Amazon Region

**DOI:** 10.3390/biomedicines12030504

**Published:** 2024-02-23

**Authors:** Marilene Ribeiro, Luis Sousa, Johnatha Oliveira, Derek Pinto, Luís Batista, Luis Lobato, Lucas Sousa, Hivylla Ferreira, Valdenice Santos, Valéria Fontes, Diana Bastos, Flor de Maria Silva, Márcio Nunes, Priscila Sabbadini, Adriana Rêgo, Amanda Aliança, Maria Silva, Washington Lima, Cláudia Lima, Mônica Gama, Lidio Lima Neto, Wellyson Firmo

**Affiliations:** 1Postgraduate Program in Management of Health Programs and Services, Campus Renascença, CEUMA University, São Luís 65075-120, Brazil; marilenefrribeiro@gmail.com (M.R.); floragyhn@gmail.com (F.d.M.S.); marcionunesbiomed@hotmail.com (M.N.); adricefs@yahoo.com.br (A.R.); profa.alianca@gmail.com (A.A.); 2Faculty of Medicine, University Hospital of Federal University of Maranhão, São Luís 65020-070, Brazil; monica.gama@ufma.br; 3Postgraduate Program in Biosciences Applied to Health, Campus Renascença, CEUMA University, São Luís 65075-120, Brazil; luis000408@ceuma.com.br (L.S.); valeria.cfontes@gmail.com (V.F.); diana_karlla@hotmail.com (D.B.); prisabbadini@gmail.com (P.S.); marirah@gmail.com (M.S.); lidio.neto@ceuma.br (L.L.N.); 4Postgraduate Program in Biodiversity and Biotechnology (BIONORTE), CEUMA University, São Luís 65075-120, Brazil; valdenicefsantos@hotmail.com (V.S.); washington001294@ceuma.com.br (W.L.); claudia001176@ceuma.com.br (C.L.); 5Health Sciences Center, State University of the Tocantina Region of Maranhão, Imperatriz 65900-000, Brazil; johnatha123321@gmail.com; 6Postgraduate Program in Health and Environment, Federal University of Maranhão, São Luís 65085-580, Brazil; ddklg377@gmail.com; 7Central Public Health Laboratory of Maranhão (LACEN-MA), São Luís 65020-904, Brazil; luisaugustosb7@gmail.com (L.B.); felipe-lobato1@hotmail.com (L.L.); lhenridss@gmail.com (L.S.); hivylla@gmail.com (H.F.); 8Postgraduate Program in Tropical Medicine, Oswaldo Cruz Institute, Rio de Janeiro 21040-360, Brazil

**Keywords:** community-acquired pneumonia, COVID-19, children, SARS-CoV-2

## Abstract

Introduction: SARS-CoV-2 infection usually presents similarly to other respiratory viral pathogens. Children and adolescents do not present as a group that is highly affected by the disease, having low infection rates. However, limited publications are associated with the findings of pneumonia in pediatric patients with COVID-19. Objective: To analyze the clinical and epidemiological aspects of children and adolescents hospitalized with SARS-CoV-2 in a pre-Amazon region. Methods: A retrospective study, carried out in four public hospitals in São Luís, Brazil where medical records of children and adolescents aged from 0 to 13 years, of both sexes, with clinical diagnosis of community-acquired pneumonia were evaluated from March 2020 to March 2021. Results: Almost 40.0% of children were aged between 1 year and 5 years. Of the 128 children who had SARS-CoV-2, 3 are of indigenous ethnicity. Additionally, 78.6% of the children had fever and there was no significant difference between COVID-19 patients and those of other respiratory viruses. Eighteen patients had chronic neurological disease, which is the most common comorbidity observed in patients with coronavirus infection. Ground glass opacity attenuation was observed in 24.8% of children and adolescents with COVID-19. Anemia and increased inflammatory response markers were related to SARS-CoV-2 infection. More than 90.0% of patients admitted to hospital, regardless of etiology, were treated with antibiotics. Eighteen patients died. Pediatric multisystem inflammatory syndrome (PMIS) was diagnosed in 17 patients. Conclusions: SARS-CoV-2 in children and adolescents is mild, but the condition of patients with PMIS is more serious, with an increase in inflammatory biomarkers which can lead to death. Therefore, rapid diagnosis and differentiation of agents causing respiratory diseases are necessary for better therapeutic decision making, since the results of this study make us question the excessive use of antibiotics without meeting well-defined clinical–epidemiological criteria.

## 1. Introduction

In March 2020, the WHO declared the sixth Public Health Emergency of International Concern—the SARS-CoV-2 pandemic [1,2]. Since then, the improvement of strategies to stop the spread and prevention of this virus has been a constant pursuit worldwide. In Brazil, the first patient of COVID-19 infection was documented in February 2020. Almost a year later, we occupied the third position in the world in accumulated cases. Brazil currently has 37,693,506 confirmed cases and 704,320 deaths, last updated in July 2023 [3]. In the state of Maranhão (MA), there have been a total of 493,620 confirmed patients and 11,55 deaths, last updated on 10 March 2023 [4].

The state of Maranhão (MA) has a population of 7 million inhabitants and represents the fourth largest state in relation to the number of inhabitants in the northeast region, corresponding to 3.4% of the Brazilian population, among which 54,121 COVID-19 cases (10.89%) were confirmed in individuals under 19 years of age. In the capital São Luís (MA), SARS-CoV-2 infection caused 2575 deaths, of which 69 (2.67%) occurred in children and adolescents up to 19 years old, with a lethality rate of 5.43% [4]. For caregivers of children and adolescents, another major challenge arose in the management of pulmonary conditions caused by this infection. Children do not present themselves as a group that is greatly affected by the disease, having low infection rates at the beginning of the pandemic [5,6]. However, social isolation may have influenced the increase in the number of cases in this age group [7].

Although COVID-19 presents a favorable clinical course in most patients, due to several factors in combination, such as the absence of previous comorbidities, low prevalence of obesity and thrombosis associated with immunological factors such as expression of the ACE-2 receptor (a viral receptor possibly involved in the infection), trained immunity and a good immune response [8], we observed that children play a relevant role in the chain of transmission of the disease, with the study of SARS-CoV-2 infection in this age group being of great importance [9]. It is worth noting that a severe presentation has been observed, probably associated with SARS-CoV-2, defined as pediatric multisystem inflammatory syndrome (PMIS), possibly due to intense macrophage activation, with clinical manifestations and laboratory changes similar to those observed in children and adolescents with complete Kawasaki syndrome, incomplete Kawasaki syndrome and/or toxic shock syndrome [10,11].

PMIS involves at least two organs and the cardiac, renal, respiratory, hematological, gastrointestinal, dermatological, or neurological systems have been described as being involved. It can occur a few days, or weeks, after acute SARS-CoV-2 infection and usually in older children (schoolchildren and adolescents), with more exuberant inflammatory markers and important elevations of cardiac injury markers [12]. In this context, our work analyzed the clinical and epidemiological aspects in a pediatric population in São Luís (MA) with SARS-CoV-2 infection for a better understanding of these pulmonary conditions. It is important to characterize the way this disease presents itself in this age group in order to recognize characteristics with potential for severity, which allow us to establish early and more assertive therapeutic approaches, which reduce the number of hospitalizations and deaths.

## 2. Materials and Methods

### 2.1. Study Design, Sites and Participants

This study was approved by the Research Ethics Committee of Ceuma University under the Certificate of Submission for Ethical Consideration (Number 3.542.361, CAAE: 20028313.3.0000.5084, approved on 29 August 2019) and was conducted in compliance with the Declaration of Helsinki. An informed consent form was not applied because the study was conducted retrospectively.

This is a cross-sectional study in infants and adolescents up to 13 years old, from March 2020 to March 2021, admitted to four public referral hospitals in São Luís (MA) with acute respiratory symptoms, of which tachypnea was adjusted for age and defined as follows: ≥60 cycles/min for newborns; ≥50/min for ages between 1 and 12 months; ≥40/min for >1 year, according to the criteria of the World Health Organization [13].

The etiology of the patients’ viral infections was identified after collection of a nasopharyngeal swab by trained personnel and subsequent extraction of viral ribonucleic acid and detection by real-time polymerase chain reaction (RT-PCR) of the collected samples in a reference laboratory of clinical analysis in São Luís (MA). The viruses identified in the viral panel were: adenovirus, respiratory syncytial virus, metapneumovirus, human rhinovirus, influenza A and B viruses and COVID-19.

The data were from the patients’ medical records and recorded on a form with multiple-choice questions specific to this study. This form included sociodemographic data, clinical signs and symptoms, laboratory and imaging diagnostic tests, pharmacological or non-pharmacological treatment used and case evolution (outcome). The forms were completed by the researchers with the information recorded in medical records of the four referral services for hospitalization of children in São Luís (MA). We emphasize that, due to the pandemic, all hospitalized study participants were tested for SARS-CoV-2.

### 2.2. Criteria for Inclusion and Exclusion

Inclusion criteria: children and adolescents of all races/ethnicities up to 13 years of age with positive nasopharyngeal tolerance samples for viral infections by molecular test, who have a record of symptoms of acute infections reported and/or observed at the time of admission. Exclusion criteria: children and adolescents who did not present flu-like symptoms, were not admitted to reference hospitals and who did not undergo molecular tests for respiratory viruses.

### 2.3. Statistical Analysis

The results obtained were analyzed using SigmaStat v.2.0 (SPSS Inc., Chicago, IL, USA). Descriptively, the absolute and percentage frequencies of the data were calculated. The significance value of *p* < 0.05 and the 95% confidence interval (CI) were used to prove statistical relevance; Pearson’s correlation coefficient was used to determine correlation between variables with normal distribution, and Spearman’s correlation was used to determine correlation between variables without normal distribution.

## 3. Results

Table 1 shows the association between the variables macro-region of origin (*p* = 0.002), hospital of origin and age group (*p* = 0.000) and COVID-19.

Table 2 shows that in relation to signs and symptoms, those that were significantly associated with COVID-19 patients were: cough, nasal obstruction, eupneic, respiratory rate, dyspnea (*p* = 0.000), pulmonary rales, wheezing, oxygen saturation (*p* = 0.001), intercostal draft (*p* = 0.002), respiratory distress (*p* = 0.005), moaning (*p* = 0.014) and coryza (*p* = 0.018).

Regarding comorbidities presented by patients and the association with COVID-19, it is noted in Table 3 that only neuropathy (*p* = 0.025) and genetic disease (*p* = 0.013) were significant.

Table 4 shows the association of imaging tests with COVID-19, with chest computed tomography (*p* = 0.000) and echocardiogram (*p* = 0.028) being significant.

Table 5 shows the association of laboratory tests with COVID-19, and those that showed significance were: hemoglobin (*p* = 0.003), Pro BNP and aspartate aminotransferase (*p* = 0.001), erythrocyte sedimentation rate (0.032), C reactive protein (*p* = 0.011), troponin, international norms ratio, patient/control ratio, fibrinogen, ferritin, D dimer, triglycerides, urea, creatinine, alanine aminotransferase, total, direct and indirect bilirubin, creatine phosphokinase, CKMB, lactate dehydrogenase, sodium, potassium, calcium, magnesium, albumin and blood culture, all with *p* = 0.000.

Regarding the therapies used and association with COVID-19, the use of antiviral and anticoagulant (*p* = 0.000), antibiotic (*p* = 0.006), immunoglobulin (0.004) and oxygen therapy (*p* = 0.013) was significant (Table 6).

Regarding the association of COVID-19 and PMIS, it is noted that there was significance with a *p* value of 0.018 (Table 7).

## 4. Discussion

The pediatric age group with SARS-CoV-2 presents individuality in its clinical characteristics, which seems to evade the notorious severity of adult and elderly patients [14,15]. In children and adolescents, a milder pattern of COVID-19 is observed, with few reports of severity when compared the adult population [16]. We identified that all the studied range is susceptible to COVID-19, but it was most observed in adolescents (45/34.9%), followed by children from 1 year to 5 years (39/30.2%) and then newborns (26/20.2%).

In our study, most children and adolescents presented respiratory symptoms similar to viral infections. Fever and cough were the most frequent symptoms. Fever and cough are the most common symptoms seen in children and adolescents with COVID-19 [17]. However, gastrointestinal, neurological and dermatological symptoms were also observed to a lesser extent [18]. It seems to us that the clinical characteristics of patients with SARS-CoV-2 are multivariate, with multisystemic involvement in their initial presentation and the lack of knowledge of the disease may have led to the devaluation of signs and symptoms other than respiratory [19].

It is emphasized that the data collection was carried out in the initial phase of the pandemic and that the lack of knowledge of the disease has resulted in a low prevalence of this symptomatology outside the respiratory context, which may have interfered with the documentation of loss of taste or smell in any hospital record, in addition to the inherent age issue. It will be important to include these signs and symptoms associated or not with respiratory conditions to suggest COVID-19 infection.

Evidence of pulmonary infiltrate on chest X-ray is considered a reference for the diagnosis of pneumonia, in addition to aiding in the detection of complications and evolutionary assessment of this pathology [20]. Most of our patients underwent chest X-ray, without defined diagnostic criteria of severity or approach to complications. This fact may be related to the care of infants, children and adolescents having occurred in an emergency hospital before hospitalization, inducing the care team to look for an imaging finding that indicated pulmonary involvement in SARS-CoV-2 in the face of a pandemic situation.

We showed that there was no association between children with COVID-19 and chest X-ray images, in agreement with several studies that showed the low sensitivity of this examination for the diagnosis of COVID-19, which may be caused by asymptomatic or mild patients or the lack of lung involvement at the time of the examination [20,21]. However, the changes observed in the chest radiographs were read by non-radiologists, which may also have influenced the definitive report and underestimated this correlation.

We evidenced that chest computed tomography showed a strong association with COVID-19, a fact already observed by several authors, who noted the greater sensitivity of this imaging exam, reaching more than 90.0%, in addition to a more defined characterization of the images in patients positive for COVID-19, having been indicated in the initial phase of the disease [22].

Approximately a quarter of patients with signs of severity underwent this examination. We report that it was not accessible for all patients, partly because the hospital does not have a computed tomography scanner to perform the examination, which required displacement to another treatment unit, resulting in risky displacement for severe or very severe patients. It is worth mentioning the importance of computed tomography in the evolutionary follow-up of pneumonia in patients positive with RT-PCR for COVID-19, aiming at the search for other differential diagnoses, in addition to the analysis of the degree of pulmonary involvement, which was lost in our evaluation [21,23].

Our work reveals that in the blood count the granulocytic, lymphocytic and platelet series may be increased or decreased, variables already identified in other studies [9,15,23]. A strong association was seen in anemic patients. Although cases of COVID-19 infection present a milder clinical picture in children and adolescents [24], some patients require intensive care assistance, as was described in patients who developed multisystem inflammatory disease during the evolution of this infection. The inflammatory response appears similar to Kawasaki disease and toxic shock syndrome [25].

Some concepts have been defined with characteristics of this persistent inflammatory process, such as: absence of identified microbial infection, organ dysfunction and positive or negative RT-PCR for COVID-19 or contact with COVID-19-positive people [26]. Of the patients described in our study, 17 (7.98%) presented conceptual diagnostic criteria of PMIS in the course of COVID-19 infection, all with severe respiratory symptoms and consequently hospitalized in the pediatric intensive care unit and treated with broad-spectrum antibiotic therapy.

Cardiac involvement in patients who had PMIS was present in more than half of the cases, showing clinical manifestation in the heart, corroborating the frequent use of inotropes throughout hospitalization [14]. Because the work was retrospective, we were unable to assess the myocardial dysfunction of children and adolescents, however, there is a record of death from cardiogenic shock, signaling the degree of inflammatory response in the cardiovascular system and the need for follow-up to identify the possible implications of COVID-19 in this system.

Coagulopathy is a significant feature in patients with PMIS, regardless of age [27]. Two of our patients died from pulmonary hemorrhage and one newborn from central nervous system bleeding. There is no report of thrombosis, however, the vast majority were treated with anticoagulation therapy based on the high D dimer value. The use of anticoagulation therapy is recommended in critically ill patients with COVID-19, but the pediatric age group carries frequent doubts about the use or non-use of this therapy, which needs to be better defined among hematologists due to the risk of complications inherent to anticoagulants, such as bleeding. We emphasize that more than half of our patients presented cardiac dysfunction and the use of acetylsalicylic acid was recommended. This association may have resulted in a higher risk of bleeding in these children who presented bleeding.

## 5. Conclusions

We concluded in this study that SARS-CoV-2 affects children and adolescents, with milder symptoms compared to severe presentations of the disease, however, when severe conditions arise they require intensive therapy, for example, PMIS, which increases inflammatory biomarkers and can lead to death. The radiological diagnosis of COVID-19 was best defined by chest computed tomography, with the finding of ground glass opacity and lung attenuation. The viral etiology identified in our analysis of hospitalized patients makes us question the excessive use of antibiotics without meeting well-defined clinical–epidemiological criteria.

## Figures and Tables

**Table 1 biomedicines-12-00504-t001:** Distribution of the characteristics related to the origin, social and demographic profile of patients according to the group and the *p*-value of the chi-square or Fisher test.

Variables	COVID-19		*p*-Value
	No	Yes	
	N (%)	N (%)	
**Macro-region of origin**			0.002
São Luís	66 (78.6)	68 (52.7)	
Caxias	00 (0.0)	06 (4.7)	
Pinheiro	07 (8.3)	13 (10.1)	
Imperatriz	01 (1.2)	02 (1.6)	
President Dutra	00 (0.0)	13 (10.1)	
Coroatá	04 (4.8)	13 (10.1)	
Santa Inês	06 (7.1)	14 (10.9)	
**Home hospital**			0.000
Children’s Hospital	69 (82.1)	32 (24.8)	
Federal University of Maranhão Hospital	12 (14.3)	24 (18.6)	
Maternal and Child Hospital Complex of Maranhão	00 (0.0)	42 (32.6)	
Carlos Macieira Hospital	03 (3.6)	31 (24.0)	
**Gender**			0.232
Female	43 (51.2)	54 (41.9)	
Male	41 (48.8)	75 (58.1)	
**Age group**			0.000
Newborn	09 (10.7)	26 (20.2)	
29 days to ˂2 months	01 (1.2)	02 (1.6)	
2 months to 11 months	34 (40.5)	17 (13.2)	
1 year to 5 years	33 (39.3)	39 (30.2)	
* Adolescent	07 (8.3)	45 (34.9)	
**Indigenous**			0.280
Yes	00 (0.0)	03 (2.3)	
No	84 (100.0)	126 (97.7)	

Abbreviations: N = number; (%) = percentage; * Adolescent = patients aged 12 and 13 years were included.

**Table 2 biomedicines-12-00504-t002:** Distribution of the signs and symptoms presented by patients according to the group and the *p*-value of the chi-square or Fisher test.

Variables	COVID-19		*p*-Value
	No	Yes	
	N (%)	N (%)	
**Fever**			0.039
Yes	66 (78.6)	83 (64.3)	
No	18 (21.4)	46 (35.7)	
**Cough**			0.000
Yes	77 (91.7)	71 (55.0)	
No	07 (8.3)	58 (45.0)	
**Cyanosis**			0.402
Yes	06 (7.1)	15 (11.6)	
No	78 (92.9)	114 (88.4)	
**Vomiting**			0.175
Yes	31 (36.9)	35 (27.1)	
No	53 (63.1)	94 (72.9)	
**Abdominal pain**			0.077
Yes	07 (8.3)	23 (18.0)	
No	77 (91.7)	105 (82.0)	
**Respiratory distress**			0.005
Yes	53 (63.1)	55 (42.6)	
No	31 (36.9)	74 (57.4)	
**Odinophagia**			0.253
Yes	03 (3.6)	11 (8.5)	
No	81 (96.4)	118 (91.5)	
**Runny nose**			0.018
Yes	36 (42.9)	34 (26.4)	
No	48 (57.1)	95 (73.6)	
**Gemency**			0.014
Yes	14 (16.7)	07 (5.4)	
No	70 (83.3)	122 (94.6)	
**Diarrhea**			0.104
Yes	04 (4.8)	16 (12.4)	
No	80 (95.2)	113 (87.6)	
**Irritability**			0.078
Yes	13 (15.5)	09 (7.0)	
No	71 (84.5)	120 (93.0)	
**Nasal obstruction**			0.000
Yes	26 (31.0)	09 (7.0)	
No	58 (69.0)	120 (93.0)	
**Lung stertors**			0.001
Yes	48 (57.1)	42 (32.6)	
No	36 (42.9)	87 (67.4)	
**Wheezing**			0.001
Yes	27 (32.1)	16 (12.4)	
No	57 (67.9)	113 (87.6)	
**Eupneic**			0.000
Yes	20 (23.8)	69 (53.5)	
No	64 (76.2)	59 (45.7)	
Not informed/realized	00 (0.0)	01 (0.8)	
**Oxygen saturation**			0.001
>92%	19 (22.6)	54 (41.9)	
˂92%	18 (21.4)	35 (27.1)	
Not informed/realized	47 (56.0)	40 (31.0)	
**Tachypnea**			0.026
Yes	36 (42.9)	35 (27.1)	
No	48 (57.1)	94 (72.9)	
**Respiratory rate**			0.000
Increased	15 (17.9)	21 (16.3)	
Normal	04 (4.8)	33 (25.6)	
Not informed/realized	65 (77.4)	75 (58.1)	
**Dyspnea**			0.000
Yes	53 (63.1)	48 (37.2)	
No	31 (36.9)	81 (62.8)	
**Nasal wing beat**			0.321
Yes	16 (19.0)	18 (14.0)	
No	68 (81.0)	111 (86.0)	
**Intercostal stripping**			0.002
Yes	39 (46.4)	33 (25.6)	
No	45 (53.6)	96 (74.4)	

Abbreviations: N = number; (%) = percentage.

**Table 3 biomedicines-12-00504-t003:** Distribution of comorbidities presented by patients, according to the group, and the *p*-value of the chi-square or Fisher test.

Variables	COVID-19		*p*-Value
	No	Yes	
	N (%)	N (%)	
**Down Syndrome**			1.000
Yes	01 (1.2)	02 (1.6)	
No	83 (98.8)	127 (98.4)	
**Asthma**			0.488
Yes	02 (2.4)	07 (5.4)	
No	82 (97.6)	122 (94.6)	
**Infectious–parasitic diseases**			0.581
No	80 (95.2)	121 (93.8)	
Visceral leishmaniasis	01 (1.2)	05 (3.9)	
Syphilis	01 (1.2)	01 (0.8)	
Human immunodeficiency virus	02 (2.4)	01 (0.8)	
Dengue	00 (0.0)	01 (0.8)	
**Heart disease**			0.580
Yes	07 (8.3)	07 (5.4)	
No	77 (91.7)	122 (94.6)	
**Hepatopathy**			1.000
Yes	00 (0.0)	01 (0.8)	
No	84 (100.0)	128 (99.2)	
**Prematurity**			0.797
Yes	08 (9.5)	15 (11.6)	
No	76 (90.5)	114 (88.4)	
**Chronic kidney disease**			0.093
Yes	01 (1.2)	09 (7.0)	
No	83 (98.8)	120 (93.0)	
**Neuropathy**			0.025
Yes	03 (3.6)	18 (14.0)	
No	81 (96.4)	111 (86.0)	
**Hematopathy**			0.154
Yes	02 (2.4)	00 (0.0)	
No	82 (97.6)	129 (100.0)	
**Genetic disease**			0.013
Yes	00 (0.0)	09 (7.0)	
No	84 (100.0)	120 (93.0)	

Abbreviations: N = number; (%) = percentage.

**Table 4 biomedicines-12-00504-t004:** Distribution of imaging tests performed on patients and alterations according to group and *p*-value of the chi-square or Fisher test.

Variables	COVID-19		*p*-Value
	No	Yes	
	N (%)	N (%)	
**Chest X-ray**			0.708
Normal	13 (15.5)	23 (17.8)	
Infiltrated/condensation	32 (38.1)	53 (41.1)	
Pleural effusion	06 (7.1)	08 (6.2)	
Other changes	01 (1.2)	05 (3.9)	
Not informed/realized	32 (38.1)	40 (31.0)	
**Chest computed tomography**			0.000
Normal	03 (3.6)	10 (7.8)	
Ground glass	01 (1.2)	32 (24.8)	
Pleural effusion	04 (4.8)	06 (4.7)	
Infiltrated/condensation	03 (3.6)	08 (6.2)	
Other changes	01 (1.2)	02 (1.6)	
Not informed/realized	72 (85.7)	71 (55.0)	
**Echocardiogram**			0.028
Amended	05 (6.0)	12 (9.3)	
Normal	02 (2.4)	15 (11.6)	
Not informed/realized	77 (91.7)	102 (79.1)	

Abbreviations: N = number; (%) = percentage.

**Table 5 biomedicines-12-00504-t005:** Distribution of laboratory tests on patients according to group and *p*-value of the chi-square or Fisher test.

Variables	COVID-19		*p*-Value
	No	Yes	
	N (%)	N (%)	
**Hemoglobin (g/dL)**			0.034
˂11 (anemia)	27 (32.1)	62 (48.1)	
11–14 (normal)	49 (58.3)	62 (48.1)	
Not informed/accomplished	08 (9.5)	05 (3.9)	
**Leukocytes (mm)^3^**			0.333
˂4000 (leukopenia)	01 (1.2)	04 (3.1)	
4000–10,000 (normal)	30 (35.7)	52 (40.3)	
>10,000 (leukocytosis)	45 (53.6)	68 (52.7)	
Not informed/realized	08 (9.5)	05 (3.9)	
**Lymphocytes (mm)^3^**			0.082
˂800 (lymphopenia)	03 (3.6)	03 (2.3)	
800–4000 (normal)	42 (50.0)	85 (65.9)	
>4000 (lymphocytosis)	31 (36.9)	36 (27.9)	
Not informed/realized	08 (9.5)	05 (3.9)	
**Neutrophils (mm)^3^**			0.313
˂1800 (neutropenia)	07 (8.3)	09 (7.0)	
1800–8000 (normal)	45 (53.6)	69 (53.5)	
>8000 (neutrophilia)	24 (28.6)	46 (35.7)	
Not informed/realized	08 (9.5)	05 (3.9)	
**Platelets (mm)^3^**			0.317
˂140,000 (thrombocytopenia)	10 (11.9)	21 (16.3)	
140,000–450,000 (normal)	50 (59.5)	81 (62.8)	
>450,000 (plateletosis)	16 (19.0)	22 (17.1)	
Not informed/realized	08 (9.5)	05 (3.9)	
**Troponin (pg/nL)**			0.000
>0.06 (increased)	03 (3.6)	22 (17.1)	
<0.06 (normal)	00 (0.0)	15 (11.6)	
Not informed/realized	81 (96.4)	92 (71.3)	
**Pro BNP**			0.001
>100 (increased)	00 (0.0)	13(10.1)	
˂100 (normal)	02 (2.4)	11 (8.5)	
Not informed/accomplished	82 (97.6)	105 (81.4)	
**International norms ratio**			0.000
>1.25 (extended)	09 (10.7)	49 (38.0)	
Up to 1.25 (normal)	18 (21.4)	43 (33.3)	
Not informed/realized	57 (67.9)	37 (28.7)	
**Patient/control relationship**			0.000
>1.25 (extended)	08 (9.5)	26 (20.2)	
Up to 1.25 (normal)	21 (25.0)	65 (50.4)	
Not informed/realized	55 (65.5)	38 (29.5)	
**Fibrinogen (mg/dL)**			0.000
˂180 (hypofibrinogenemia)	01 (1.2)	12 (9.3)	
180–350 (normal)	02 (2.4)	26 (20.2)	
>350 (hyperfibrinogenemia)	01 (1.2)	14 (10.9)	
Not informed/realized	80 (95.2)	77 (59.7)	
**Ferritin (ng/nL)**			0.000
>322 (increased)	03 (3.6)	21 (16.3)	
<22–322 (normal)	03 (3.6)	35 (27.1)	
Not informed/realized	78 (92.9)	73 (56.6)	
**Erythrocyte sedimentation rate**			0.032
>10 (increased)	02 (2.4)	11 (8.5)	
Up to 10 (normal)	01 (1.2)	08 (6.2)	
Not informed/realized	81 (96.4)	110 (85.3)	
**D dimer (ng/nL)**			0.000
>500 (increased)	02 (2.4)	52 (40.3)	
Up to 500 (normal)	02 (2.4)	09 (7.0)	
Not informed/realized	80 (95.2)	68 (52.7)	
**Triglycerides (mg/dL)**			0.000
>150 (increased)	04 (4.8)	19 (14.7)	
˂150 (normal)	05 (6.0)	28 (21.7)	
Not informed/realized	75 (89.3)	82 (63.6)	
**C reactive protein (mg/dL)**			0.011
>0.04 (increased)	25 (29.8)	63 (48.8)	
˂0.04 (normal)	21 (25.0)	30 (23.3)	
Not informed/realized >39 (increased)	38 (45.2) 06 (7.1)	36 (27.9) 28 (21.7)	
**Urea (mg/dL)**			0.000
15–39 (normal)	35 (41.7)	71 (55.0)	
Not informed/realized	43 (51.2)	30 (23.3)	
**Creatinine (mg/dL)**			0.000
>1.3 (increased)	03 (3.6)	14 (10.9)	
0.55–1.3 (normal)	19 (22.6)	74 (57.4)	
Not informed/realized	62 (73.8)	41 (31.8)	
**Aspartate aminotransferase (U/L)**			0.001
>37 (increased)	16 (19.0)	42 (32.6)	
˂37 (normal)	26 (31.0)	55 (42.6)	
Not informed/realized	42 (50.0)	32 (24.8)	
**Alanine aminotransferase (U/L)**			0.000
>63 (increased)	02 (2.4)	12 (9.3)	
˂63 (normal)	34 (40.5)	84 (65.1)	
Not informed/realized	48 (57.1)	33 (25.6)	
**Alkaline phosphatase (U/L)**			0.114
>136 (increased)	10 (11.9)	25 (19.4)	
˂136 (normal)	01 (1.2)	06 (4.7)	
Not informed/realized	73 (86.9)	98 (76.0)	
**Total bilirubin (mg/dL)**			0.000
>1 (increased)	03 (3.6)	16 (12.4)	
˂1 (normal)	10 (11.9)	51 (39.5)	
Not informed/realized	71 (84.5)	62 (48.1)	
**Direct bilirubin (mg/dL)**			0.000
>0.2 (increased)	07 (8.3)	20 (15.5)	
˂0.2 (normal)	06 (7.1)	47 (36.4)	
Not informed/accomplished	71 (84.5)	62 (48.1)	
**Indirect bilirubin (mg/dL)**			0.000
>0.8 (increased)	01 (1.2)	13 (10.1)	
<0.8 (normal)	12 (14.3)	54 (41.9)	
Not informed/realized	71 (84.5)	62 (48.1)	
**Creatine phosphokinase (U/L)**			0.000
>225 (increased)	05 (6.0)	11 (8.5)	
<225 (normal)	06 (7.2)	46 (35.7)	
Not informed/realized	72 (86.7)	72 (55.8)	
**CKMB (ng/mL)**			0.000
>6.36 (increased)	01 (1.2)	15 (11.6)	
<6.36 (normal)	02 (2.4)	29 (22.5)	
Not informed/realized	81 (96.4)	85 (65.9)	
**Lactate dehydrogenase (IU/L)**			0.000
Increased	03 (3.6)	21 (16.3)	
Normal	06 (7.1)	26 (20.2)	
Not informed/realized)	75 (89.3)	82 (63.6)	
**Sodium (mEq/L)**			0.000
<136 (hyponatremia)	07 (8.3)	25 (19.4)	
>145 (hypernatremia)	19 (22.6)	71 (55.0)	
136–145 (normal)	03 (3.6)	07 (5.4)	
Not informed/realized	55 (65.5)	26 (20.2)	
**Potassium (mEq/L)**			0.000
<3.5 (hypopotassemia)	02 (2.4)	10 (7.8)	
>5 (hyperpotassemia)	22 (26.2)	80 (62.0)	
3.5–5 (normal)	03 (3.6)	13 (10.1)	
Not informed/realized	57 (67.9)	26 (20.2)	
**Calcium (mg/dL)**			0.000
<8 = hypocalcemia	04 (4.8)	16 (12.4)	
>10 = hypercalcemia	26 (31.0)	68 (52.7)	
8–10 = normal	01 (1.2)	04 (3.1)	
Not informed/realized	53 (63.1)	41 (31.8)	
**Magnesium (mg/dL)**			0.000
>2.6 (increased)	00 (0.0)	05 (3.9)	
1.6–2.6 (normal)	20 (23.8)	79 (61.2)	
Not informed/realized	64 (76.2)	45 (34.9)	
**Albumin (g/dL)**			0.000
<3.4 (hypoalbuminemia)	06 (7.1)	25 (19.4)	
3.4–5 (normal)	07 (8.3)	36 (27.9)	
Not informed/realized	71 (84.5)	68 (52.7)	
**Blood culture**			0.000
Yes	02 (2.4)	14 (10.9)	
No	13 (15.5)	51 (39.5)	
Not informed/realized	69 (82.1)	64 (49.6)	

Abbreviations: N = number; (%) = percentage.

**Table 6 biomedicines-12-00504-t006:** Distribution of therapy used on patients according to group and *p*-value of the chi-square or Fisher test.

Variables	COVID-19		*p*-Value
	No	Yes	
	N (%)	N (%)	
**Antiviral**			0.000
Yes	48 (57.1)	29 (22.5)	
No	36 (42.9)	100 (77.5)	
**Antibiotic**			0.006
Yes	81 (96.4)	107 (82.9)	
No	03 (3.6)	22 (17.1)	
**Corticoids**			0.101
Yes	59 (70.2)	75 (58.1)	
No	25 (29.8)	54 (41.9)	
**Immunoglobulin**			0.004
Yes	00 (0.0)	11 (8.5)	
No	84 (100.0)	118 (91.5)	
**Anticoagulant**			0.000
Yes	00 (0.0)	19 (14.7)	
No	84 (100.0)	110 (85.3)	
**Dialysis**			1.000
Yes	02 (2.4)	03 (2.3)	
No	82 (97.6)	126 (97.7)	
**Oxygen therapy**			0.013
Non-invasive ventilation	34 (40.5)	39 (30.2)	
No	45 (53.6)	64 (49.6)	
Invasive ventilation	05 (6.0)	26 (20.2)	

Abbreviations: N = number; (%) = percentage.

**Table 7 biomedicines-12-00504-t007:** Distribution by patients presenting pediatric multisystem inflammatory syndrome signs and symptoms according to group and *p*-value of Fisher test.

Variables	COVID-19		*p*-Value
	No	Yes	
	N (%)	N (%)	
**Pediatric multisystem inflammatory syndrome**			0.018
Yes	02 (2.4)	15 (11.6)	
No	82 (97.6)	114 (88.4)	
**Skin rash**			0.407
Yes	01 (1.2)	05 (3.9)	
No	83 (98.8)	124 (96.1)	
**Conjunctivitis**			0.093
Yes	01 (1.2)	09 (7.0)	
No	83 (98.8)	120 (93.0)	
**Lesion in oral cavity**			1.000
Yes	01 (1.2)	03 (2.3)	
No	83 (98.8)	126 (97.7)	
**Bleeding**			1.000
Yes	03 (3.6)	05 (3.9)	
No	81 (96.4)	124 (96.1)	

Abbreviations: N = number; (%) = percentage.

## Data Availability

Data are available and will be sent by email upon request.

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
