# Peer review of "Clinical and Epidemiological Assessment of Children and Adolescents Hospitalized with SARS-CoV-2 in the Pre-Amazon Region"

_biomedicines, 2024, doi:10.3390/biomedicines12030504_

Round 1

Reviewer 1 Report

Comments and Suggestions for Authors

In the MS biomedicines-2855319, the authors attract the readers' attention with a retrospective study regarding "CLINICAL AND EPIDEMIOLOGICAL EVALUATION OF HOSPITALIZED CHILDREN WITH SARS-COV-2 IN A CITY OF PRE-AMAZON REGION."

Despite being retrospective, the study appears interesting because Coronavirus still has a high frequency in people of all ages, especially in cold weather, inducing various clinical symptoms.

The following comments and suggestions are available below.

1. Major

Abstract:

Lines 18-21: Please separate the long phrase into 2-3 parts for better understanding;

Lines 26-27:  "Three indigenous patients had SARS-CoV-2." Did exist other ones, non-indigenous? If the response is positive, please explain clearly what the term indigenous means in the current MS.

Line 30: Does "ground glass" mean ground glass opacity (GGO) as a term used in lung radiological examination?

Lines 33-34: SIM-P was diagnosed in 17 patients. Conclusions: Respiratory viruses were the main etiological agents in CAP. Please show the significance of SIM-P and CAP.

Introduction

Lines 47-52: the authors present data about people up to 19 years of age, and in the study, they selected children and adolescents aged from zero to 13 years (abstract, line 24). In lines 130-134, they did not indicate the age of the recruited group. Please check, correct, and/or explain.

Lines 66 and 70: is PMIS the same as SIM-P? If the response is positive, the authors should put only one abbreviation in the MS text for more clarity.

Materials and Methods: 

In this section, the authors included several subsections, as follows: 

2.1. Study design, sites and participants

The authors show the study period (March 2020- March 2021 - 1 year) age of the children (0-13 years) admitted to the hospital (four public referral hospitals in São Luís-MA) with acute respiratory symptoms. They show   tachypnea values adjusted for age and defined as follows: ≥60 cycles/min for newborns; ≥50/min for ages between 1 and 12 months; ≥40/min 85 for >1 year), according to the criteria of the World Health Organization. 

They indicate that the viral etiology was evaluated using RT-PCR, and the viruses identified in the viral panel were adenovirus, Respiratory Syncytial Virus, metapneumovirus, human rhinovirus, influenza A and B viruses, and COVID-19.

The authors indicated that Data collection was recorded on a form with multiple-choice questions suitable for this study. This form included socio-demographic data, clinical signs and symptoms, diagnostic tests, treatment used, and outcome. The researchers completed the forms with the information recorded in the medical records of the four referral services.

Due to the pandemic, they emphasize that all hospitalized study participants were tested for SARS-CoV-2.

2.2. Extraction of genetic material

2.3. Molecular analysis

These subsections describe the materials and methods used for pathogen virus identification. For all viruses, or only for Coronavirus? The authors did not specify.

The reviewer believes that they are irrelevant to this study; many laboratory analyses were found in the results. 

The authors are invited to show the provenance of the apparatus, reagents, and software (Company name, City, State, and Country) used in this study.

2.4. Statistical analysis

2.5. Ethical aspects - are not materials and methods

2.6. Criteria for inclusion and exclusion 

Lines 129-134: the authors are encouraged to provide more details regarding the Criteria for inclusion and exclusion: age, zone, notations as indigenous or not, adolescent, etc. Moreover, they are invited to mention the total number of children recruited in the present retrospective study because in "Results," the authors showed the number and calculated the percentage. 

Results: 

While Section 2 (Materials and Methods) has several subsections, Section 3 (Results) has only one,  3.1. Clinical and Epidemiological Data

Numerous results are displayed in Tables; the authors used several abbreviations (unexplained in the Table footer). 

The reader is very little informed about all these data (recorded in 7 Tables) as follows: "Data collection was recorded on a form with multiple-choice questions suitable for this study. This form included socio-demographic data, clinical signs and symptoms, diagnostic tests, treatment used, and outcome. The researchers completed the forms with the information recorded in the medical records of the four referral services."

They are encouraged to show more data in Materials and Methods because the data registered are not without reason. They are invited to justify why these parameters were selected, why all analyses were performed, and what symptoms the children had when admitted to the hospital reported to the admission criteria. If the children were COVID-negative, what virus was identified? Did the authors include children of 5-13 years old in the adolescent group? Why was the reason for therapeutic protocol? 

The results included in all Tables need detailed interpretations and observations.

The authors are encouraged to organize their MS data based on all the mentioned comments. 

A descriptive statistic or, if possible, an analysis of correlations between variable parameters is required to support the Discussion.

Moreover, the authors are encouraged to reformulate the Conclusions of their study. In the current version, they are too general. 

2. Minor

Please edit the references in MDPI style.

The authors are invited to rigorously use English editing software to extensively correct the entire MS text. This is a common procedure for all authors who are not native English speakers. 

Author Response

REVIEWER 1

Author descriptions have been updated

Summary

Lines 18-21: The introduction has been improved, separating it into 3 smaller paragraphs

Lines 26-27: In relation to indigenous children, we decided to highlight that of the children affected by COVID-19, three are indigenous, a vulnerable group in our country.

Line 30: Yes, it is ground glass opacity (GGO).

Lines 33-34: rewritten leaving only PSMI and COVID-19

Introduction

Lines 47 to 52: In relation to this paragraph, the data refer to another study, specifically the city of São Luís, while our study was carried out with children and adolescents from other municipalities, in addition to limiting only up to 13 years of age. age.

Lines 66 and 70: corrected and standardized the acronym for PMIS

Materials and methods:

2.2. Extraction of genetic material

2.3. Molecular analysis

The authors chose to remove these sections, since these analyzes were not carried out in the work, it was just the intention to point out how the reference laboratory responsible for the diagnosis and identification of the viruses performed the technique, this improves the description of the data in the results as well, as it points out only epidemiological and clinical data. Therefore, follow the instructions for removing the text.

Furthermore, there is no need to add questions about equipment and reagents.

2.6. Inclusion and exclusion criteria

Lines 129-134: The authors rewrote

Results:

The authors reorganized the methodology to follow the same sequence and characteristics of the results.

Regarding the abbreviations, they were all inserted in the footer, and others were placed in the table itself, the abbreviations related to the measurements, especially in table 5, the authors thought it best not to include the meaning, since the measurements are universal, but we are open to suggestions.

The writing of the methodology to improve the collected data was carried out, explaining the collected data, justifying the children's choice in the criteria.

Regarding questions about age, there were no children aged 6 to 11, only 12 to 13 years old, which we included as teenagers, as we stratified by age group, we think it is pertinent to leave teenagers, we are open to suggestions.

84 children/adolescents were not positive for COVID-19, however, 62 were positive for other respiratory viruses, 57 for Respiratory syncytial virus and 5 for human rhinovirus.

The conclusion has been rewritten.

Reviewer 2 Report

Comments and Suggestions for Authors

The current study titled “CLINICAL AND EPIDEMIOLOGICAL EVALUATION OF 2 HOSPITALIZED CHILDREN WITH SARS-COV-2 IN A CITY 3 OF PRE-AMAZON REGION” Ref: 2855319, considered clinical study for limited number of 13 years old children, (March 2020 - March 2021). Several laboratory tests were collected and analyzed. Intensive discussions were mentioned due to the collected observations in understandable language. Minor revisions are needed considering the following items.

- Conclusions due to this study should be revised with well explained/definite point supported by observations.

- List of abbreviations is needed.

Author Response

REVIEWER 2

The conclusion was rewritten, as well as the abbreviations throughout the text.

Reviewer 3 Report

Comments and Suggestions for Authors

Dear Editor

The manuscript title is scientifically sound “CLINICAL AND EPIDEMIOLOGICAL EVALUATION OF HOSPITALIZED CHILDREN WITH SARS-COV-2 IN A CITY OF PRE-AMAZON REGION”, however there are several queries:

1. The objectives which are described in abstract do not match with title.

2. Title needs revision as “CLINICAL AND EPIDEMIOLOGICAL EVALUATION OF HOSPITALIZED CHILDREN WITH SARS-COV-2 IN PRE-AMAZON REGION”. Please see the comment 6 as well.

3. There are several mistakes in the text regarding “Basic English”. Line 18-20 and 21-23 needs focused wording regarding the Introduction and Objectives of the study.

4. Line 23-26: Methods section is poorly described. The authors could include RT-PCR statement here.

5. Line 27: “Three indigenous patients had SARS-CoV-2”. This is not clear to understand. The authors include the patients with SARS-C0V-2??

6. Line 33-34: Conclusion of the Authors seems to modify the title and to describe “Epidemiological Evaluation of SARS-Cov-2 and CAP”.

7. The authors are suggested to use one style for “pediatric multisystem inflammatory syndrome (PMIS) Line 66 or SIM-P as in Line 35.

8. The introduction and Methods section is written scientifically sound.

9. The results need little clarity regarding the “written explanations of the table”. It is suggested to reduce the text i.e. Line 138 “(p=0.002), hospital of origin (p=0.000) and age group (p=0.000) with COVID-19 138 cases”

10. There are long paragraphs in “Discussion section” without any references of interpretation i.e. Lines 190-198.

The discussion section needs revisions regarding “the statements of previously cited articles”.

Thanks and Regards

Comments on the Quality of English Language

The language needs significant revisions. Regards

Author Response

REVIEWER 3

The title was modified according to the suggestion of the reviewers, also including adolescents, since there is data from this group, the objective was also organized.

The summary was rewritten, listing the title, objectives and conclusion.

Line 23-26: The authors chose to remove these sections, since these analyzes were not performed in the work, it was only the intention to point out how the reference laboratory responsible for the diagnosis and identification of the viruses performed the technique, this improves the description of the data in the results also, since it only points to epidemiological and clinical data. Therefore, follow the instructions for removing the text.

Line 27: In relation to indigenous children, we decided to highlight that of the children affected by Covid 19, three are indigenous, a vulnerable group in our country.

All children and adolescents in the group were tested for Covid-19, but not all of them had the infection.

Line 33-34: The conclusion has been rewritten

The acronym PMIS was standardized throughout the text

The description of tables in the result section has been improved.

Discussion sections have been revised for need for citation and further discussion.

Reviewer 4 Report

Comments and Suggestions for Authors

Ribeiro et al. presented a manuscript entitled Clinical and epidemiological evaluation of hospitalized children with sars-cov-2 in a city of pre-amazon region. In my opinion, the topic is original and has strong scientific value.

There are a few points that need to be addressed before the article gets accepted:

All abbreviations should be explained every first time they appear in text or abstract.

Improve the Introduction. Provide more references for this part.

The conclusion part needs to be improved.

would recommend the authors to provide more discussion.

Precise the phrase: “this organ something…” p.12, line 241

Don’t use phrases like: “I also emphasize…”, page 11

Citation: use [1], [2], etc. to number the citations.

Citation is not according to MDPI rules; any article has a DOI number.

Comments on the Quality of English Language

Minor editing of English language required

Author Response

REVIEWER 4

The introduction was revised, adding new citations, the conclusion was rewritten, based on the objectives, methodology and results, the discussion was inserted new discussions and citations. The citations followed the magazine's model, the affirmative phrases were revised.

Improve the introduction. Please provide more references for this part.

Citations and references were organized according to the MDPI rule

Round 2

Reviewer 1 Report

Comments and Suggestions for Authors

The reviewer appreciates the authors' efforts to revise their MS according to the previous review report.

However, a few comments are available below:

Table 1 footer: please mention the "adolescents" signification regarding the number of years of age.

Line 145: The authors wrote, "Distribution of comorbidities presented by the cases..." They are invited to check and correct because the patients have comorbidities (as they showed in line 143), not the cases.

Similar suggestions are available for lines 151, 162, 169, 177, 

Moreover, they are encouraged to check again and correct the entire MS. According to the concept of "patient-centered medicine," a case study/series includes patients with various diseases and investigations, not cases).

Reviewer 4 Report

Comments and Suggestions for Authors

The authors have addressed all of my concerns with the original manuscript. The revised manuscript is ready for publication.
